


# Biomarker evidence for the occurrence of anaerobic ammonium oxidation in the eastern Mediterranean Sea during Quaternary and Pliocene sapropel formation

Darci Rush[1,2], Helen M. Talbot[2*], Marcel T.J. van der Meer[1], Ellen C. Hopmans[1], Ben Douglas[2], Jaap. S. Sinninghe Damsté[1,3]

[1]*Department of Marine Microbiology and Biogeochemistry, NIOZ Royal Netherlands Institute for Sea Research, and Utrecht University, Den Burg, The Netherlands*

[2]*School of Natural and Environmental Sciences, Newcastle University, Newcastle-upon-Tyne, UK*

[3] *Department of Earth Sciences, Geochemistry, Faculty of Geosciences, Utrecht University, Utrecht, The Netherlands*

*now at Department of Archaeology (BioArCh), University of York, Heslington, York, UK*

Keywords: anaerobic ammonium oxidation, sapropel, nitrogen cycle, ladderane, bacteriohopanetetrol, BHT isomer





Abstract
The eastern Mediterranean Sea sedimentary record is characterised by intervals of
organic rich sediment (sapropels), indicating periods of severe anoxia triggered by
astronomical forcing. It has been hypothesized that nitrogen fixation was crucial in
injecting the Mediterranean Sea with bioavailable nitrogen (N) during sapropel events.
However, the evolution of the N biogeochemical cycle of sapropels is poorly
understood. For example, the role of the complementary removal reaction, anaerobic
ammonium oxidation (anammox), has not been investigated because the traditional
lipid biomarkers for anammox, ladderane fatty acids, are not stable over long periods
in the sedimentary record. The recent development of an alternative lipid biomarker
for anammox (bacteriohopanetetrol stereoisomer; BHT isomer) allowed for the
investigation of anammox during sapropel deposition in this marginal sea. We present
here the first application of a lipid biomarker for N removal throughout the progression
(e.g. formation, propagation, and termination) of basin-wide anoxic events. In this
study, BHT isomer and ladderanes were analysed in sapropel records taken from
three Eastern Mediterranean sediment cores, spanning the most recent (S1) to
Pliocene sapropels. Ladderanes were rapidly degraded in sediments, as recently as
the S5 sapropel (ca. 125 ka). BHT isomer, however, was present in all sapropel
sediments, as far back as the Pliocene (2.97 Ma), and clearly showed the response of
anammox bacteria to marine water column redox shifts in high-resolution records. Two
different N removal scenarios were observed in Mediterranean sapropels. During S5,
anammox experienced Black Sea-like water column conditions, with the peak of BHT
isomer coinciding with the core of the sapropel. Under the alternative scenario
observed in the Pliocene sapropel, the anammox biomarker peaked at onset and
termination of said sapropel, which may indicate sulphide inhibition of anammox during
the core of sapropel deposition. This study shows the use of BHT isomer as a
biomarker for anammox in the marine sediment record and highlights its potential in
reconstructing anammox during past anoxic events that are too old for ladderanes to
be applied (e.g. the history of oxygen minimum zone expansion and oceanic anoxic
events).



## 1. Introduction

The typical hemipelagic, carbonate-rich, organic carbon-poor sediment record of the eastern Mediterranean Sea is periodically interspersed with dark, organic-rich layers, known as sapropels. Sapropels typically have total organic carbon (TOC) content of >2%, a striking contrast to non-sapropel TOC-lean sediments in the area, with TOC contents of generally 0.2 – 0.6% (Cramp and O'Sullivan, 1999;Mobius et al., 2010). Evidence of Mediterranean sapropels can be found as far back 13.5 Ma in the sedimentary record. These features are the result of changes in astronomical forcing (Rossignol-Strick, 1983). Briefly, at maximum insolation, a wetter, localised monsoonal climate caused an increased discharge of freshwater into the Eastern Mediterranean mainly from the African continent. This brought terrestrial nutrients into the oligotrophic Eastern Basin, while at the same time forming a layer of lower salinity water at the surface of the Mediterranean, inhibiting ventilation of deeper waters (for recent review see Rohling et al., 2015). The consequence of these climate-induced changes were (1) an increase in primary productivity followed by remineralisation and increased oxygen consumption in the underlying waters, and (2) reduced resupply of oxygen to bottom waters leading to a ventilation crisis in the Mediterranean. Combined, this led to anoxia (Sinninghe Damsté and Hopmans, 2008; and euxinia during the most intense sapropel events, cf. Menzel et al., 2002), which is believed to have started first in the pore and bottom waters and progressively shoaled over hundreds of years until the Mediterranean was characterised by photic zone anoxia (euxinia). There is some dispute over whether high TOC values observed in sapropel sediments is primarily due to enhanced productivity, better preservation under anoxic conditions, or a combination of both.

The degree of oxygen depletion and presence of euxinic conditions for individual sapropels can vary according to the strength of astronomical forcing. A recent sapropel, S5 (121 – 128.5 ka), is the most well-developed Late Quaternary sapropel, characterised by high TOC content (ca. 7 – 8 %, max. 12%), low bioturbation, and evidence for photic zone euxinia (Marino et al., 2007;Rohling et al., 2006;Struck et al., 2001). In comparison, however, certain Pliocene sapropels have been shown to contain much more elevated TOC content, of up to 30% (Nijenhuis and de Lange, 2000), suggesting that sapropels from these periods are more developed. Spatial variation also occurs during a sapropel formation, with TOC-rich horizons more





commonly forming in the east of the basin, but oxygen depletion not necessarily being
stronger in the east (cf. Menzel et al., 2002).
The reorganisation of nutrient cycles, e.g. the phosphorus (P) cycle (Slomp et al.,
2004), and the nitrogen (N) cycle (Calvert et al., 1992;Higgins et al., 2010) can impact
the production and preservation of organic matter during the formation of
Mediterranean sapropels. It has been shown that the anoxic water column during
sapropel deposition caused enhanced regeneration of sedimentary P (Slomp et al.,
2002). If sporadic vertical mixing then brought P to the photic zone, this would have
further offset the Redfield ratio. The input of terrestrial N was likely insufficient to
balance the enhanced sedimentary P remineralisation that occurred in the newly
anoxic water column. This would have shifted phytoplankton communities towards
diazotrophy (Higgins et al., 2010).
It appears that under anoxic water column conditions in the Mediterranean, N might
already have been a limiting nutrient. However, N can also be removed from the
marine system via denitrification and anaerobic ammonium oxidation (anammox)
(Ward, 2013). Anammox is the oxidation of ammonium using nitrite as the electron
acceptor to produce $N_2$, and is performed by anaerobic, sulfide-sensitive (Jensen et
al., 2008), chemolithoautotrophic bacteria (Strous et al., 1999). Anammox has been
observed in the water columns of modern oxygen minimum zones (Hamersley et al.,
2007;Pitcher et al., 2011;Rush et al., 2012b), and euxinic basins (Jensen et al.,
2008;Kuypers et al., 2003;Wakeham et al., 2012). The anammox process is also
proposed to have been an important N cycling process during Cretaceous oceanic
anoxic events (Kuypers et al., 2004), removing bio-available N for primary production
and forcing a shift in the phytoplankton community to nitrogen-fixing organisms.
However, whether anammox is a positive- or negative-feedback to anoxia during
sapropel formation is poorly understood. For instance, is the removal of N from the
system a way to quench primary productivity, the main source of the organic matter
that is remineralised and consuming oxygen? Or, does anammox simply contribute to
the continuous removal of N, much in the same way it does in modern euxinic basins
like the Cariaco Basin and the Black Sea? Studying the occurrence of anammox
during the propagation of sapropels might help clarify the role anammox plays in
maintaining anoxic conditions.





The presence of anammox in water column and sediments is usually inferred from biomarker evidence of ladderane fatty acids. Ladderane lipids contain concatenated cyclobutane rings (Fig. 1) and are synthesised exclusively by anammox bacteria (Sinninghe Damsté et al., 2002). However, ladderanes are labile lipids and are known to be susceptible to diagenetic modification in the sediment record (Rush et al., 2012a;Jaeschke et al., 2008). An alternative biomarker for anammox bacteria in paleo-records has recently been proposed to be bacteriohopanetetrol isomer (BHT isomer; Fig. 1), a much less common stereoisomer of the ubiquitous BHT. Both BHT and BHT isomer are synthesised by marine anammox bacteria ('*Ca.* Scalindua sp.') in roughly equal amounts (Rush et al., 2014b). Notably, the synthesis of BHT isomer has also been seen in a few other non-anammox, non-marine bacteria (van Winden et al., 2012;Rosa-Putra et al., 2001;Peiseler and Rohmer, 1992), and, therefore, some care should be taken when applying this lipid as a biomarker for anammox. However, anammox is the only known marine source of BHT isomer, and BHT isomer has been shown to correlate with ladderanes (Rush et al., 2014b) and metagenomic evidence for anammox bacteria (Matys et al., 2017) in modern oxygen deficient marine settings.

Anammox bacteria use the carbon assimilation pathway acetyl co-enzyme A (Strous et al., 2006). This pathway has been shown to result in the production of severely depleted ladderane fatty acids, observed in both cultures and in the Black Sea water column ($\delta^{13}C$ ~ –45‰; Schouten et al., 2004). In cultures, a $C_{30}$ hopene also had similar isotopically depleted values as the ladderane fatty acids. Isotopically depleted BHT isomer ($\delta^{13}C$ value of -51‰) was detected in a singular Pliocene sapropel sample in the Ionian Basin of the eastern Mediterranean (ODP Leg 160, Site 964) (Hemingway et al., 2018). In the same sample, BHT was 21‰ more enriched than BHT isomer. These results indicate that BHT isomer observed in a Mediterranean sapropel was derived from anammox bacteria.

Three Mediterranean sapropel records were analysed for ladderanes and/or BHT isomer. Here, for the first time, we report the presence of anammox in high resolution Mediterranean sapropel records. We assess the periodic formation of anoxia in the paleorecord of a constrained basin, and discuss its potential impact on N cycling.





## 2. Method

### 2.1. Sapropel cores

#### 2.1.1. Recent S1 – S5 sapropels (Aegean Sea)

Core LC21 was collected at 1522 m water depth in the Aegean Sea (34°40'N, 26°35'E; Fig. 2) by the R/V Marion Dufresne in 1995. The split cores have been stored in the British Ocean Sediment Core Research Facility (BOSCORF) in Southampton, UK, and were subsampled in 2014 for BHT analyses. A total of 19 sediments were collected from sapropels S1, S3, S4, and S5, with a background sediment sample from outside each sapropel (taken from sections either before or after the sapropel event). Sediments were freeze-dried and stored at -20°C until extraction.

#### 2.1.2. High-resolution S5 sapropel (Levantine Basin)

An S5 sapropel (core 64PE406-E1) was sampled in relatively high resolution (1-cm slices) from a piston core taken at a water depth of 1760 m in the Eastern Basin (Station 1; 33° 18 ' N, 33° 24' E; Fig. 2) aboard the R/V Pelagia in January 2016. The core was opened and slices were immediate transferred to geochemical bags and stored at -40°C until sediments were freeze-dried in preparation for biomarker lipid extraction and bulk TOC and isotopic analyses.

#### 2.1.3. High-resolution Pliocene sapropel (Levantine Basin)

Site 967 of ODP Leg 160 was located at a water depth of 2560 m, south of Cyprus on the lower northern slope of Eratosthenes Seamount, in the Eastern Levantine Basin (34°04N, 32°33E; Fig. 2). 33 1-cm slices were selected from Hole B, Core 9, Section 6. These were from 40 – 87 cm within the core section, corresponding to absolute depths of 79.70 – 80.16 meters below sea floor (mbsf). This sample set included sediments from above, within, and below the sapropel horizon S73, which was characterised by dark coloured sediment. According to ODP Leg 160 shipboard biostratigraphic studies, the sediment for this core is of Pliocene age, 2.97 Ma (Emeis and Party, 1996). Sediment was freeze-dried and prepared for lipid extraction and TOC measurements.

### 2.2. TOC content



Ca. 0.1 g of freeze-dried sediments from LC21 and ODP 967 were weighed
individually into a porous crucible. HCl (1 mL, 4 mM) was added to remove any
inorganic carbon from the sediment. After HCl was drained, samples were neutralised
with deionised water, and were dried at 65 ºC. TOC content of each sample was
obtained by means of non-dispersive infrared spectrometry using a LECO CS230
analyser. A standard (Chinese stream sediment, NCS DC 73307; LGC, Teddington,
UK) was analysed after every 10 samples to check accuracy. TOC content of the
64PE406-E1 sediments was determined by a Thermo Scientific Flash 2000 elemental
analyser coupled to a Thermo Scientific Delta V isotope ration monitoring mass
spectrometer (EA-irMS) via a Conflo IV.
2.3. Bulk isotope measurements
Freeze dried 64PE406-E1 sediments were analyzed to determine both bulk $\delta^{15}N$ and
bulk $\delta^{13}C$ values. For carbon isotope analysis, the sediment was first decalcified using
a 2N HCL solution for approximately 18 h. The sediment was rinsed three times using
double-distilled water and then freeze-dried again. $^{15}N_{TOC}$ and $^{13}C_{TOC}$ were measured
using a Thermo Scientific EA-irMS (see above). The $^{15}N_{TOC}$ and $^{13}C_{TOC}$ are expressed
relative to air and the Vienna Pee Dee Belemnite (VPDB) standard, respectively and
the isotope analysis precision was 0.2 ‰. For nitrogen isotope analysis, acetanilide,
urea, and casein with predetermined isotope values were used as reference material;
for carbon analysis benzoic acid and acetanilide were used.

2.4. Lipid extractions
177         2.4.1. Bligh and Dyer lipid extractions

Sediments from the LC21 (Aegean Sea; S1 – S5) and ODP 967 (Levantine Basin;
Pliocene) were extracted at Newcastle University using a modified Bligh and Dyer
extraction (BDE) method (Bligh and Dyer, 1959;Cooke et al., 2008). Briefly, freeze-
dried material was extracted in a 10:5:4 (v:v:v) mixture of MeOH:chloroform:$H_2O$ in a
Teflon tube, sonicated for 15 min at 40°C, and centrifuged for 10 min. After the
supernatant was transferred to a second tube, the residue was re-extracted two more
times. The chloroform in the supernatant was separated and collected from the
aqueous phase by making $H_2O$:MeOH ratio 1:1 (v:v). This procedure was repeated for
the subsequent extractions. The collected BDE was dried by rotary evaporation in a



round-bottom flask. Lipid extraction on the high-resolution S5 sapropel (64PE406-E1;
Levantine Basin) was performed at NIOZ, where the extraction protocol was similar,
but instead used MeOH:Dichloromethane (DCM):phosphate-buffer in the solvent
mixtures (see Rush et al., 2012a). All BDE were analysed for BHT isomer, where $C_{16}$
platelet activating factor (PAF) standard (1-O-hexadecyl-2-acetyl-sn-glycero-3-
phosphocholine was added as an internal standard. Aliquots from the 64PE406-E1
BDEs were taken for ladderane extractions.

194         2.4.2. Ladderane fatty acid extractions

Sediment of LC21 were also ultrasonically extracted 3 times using a DCM/methanol
mixture (2:1 v/v). Extracts of LC21 sediments were combined and dried using rotary
evaporation yielding the total lipid extract (TLE), and residues were reserved for direct
saponification. The LC21 TLEs, residues, and the aliquots of the 64PE406-E1 BDEs
were saponified by refluxing with aqueous KOH (in 96% MeOH) for 1h. Fatty acids
were obtained by acidifying the saponified samples to a pH of 3 with 1N HCl in MeOH
and extracted using DCM. The fatty acids were converted to their corresponding fatty
acid methyl esters (FAMEs) by methylation with diazomethane. $N_2$ was not used to aid
evaporation of solvents after derivatisation as this practice was found to significantly
decrease the yield of volatile short-chain ladderane fatty acids (Rush et al., 2012a).
Instead solvents were air dried. Polyunsaturated fatty acids (PUFAs) were removed
by eluting the sample over a small AgNO3 (5%) impregnated silica column with DCM.
Fatty acid fractions were stored at 4 °C until analysis.
2.5. Lipid analyses

209         2.5.1. Analysis of derivatised BHT isomer (Newcastle University)

A known amount of internal standard (5α-pregnane-3β,20β-diol) was added to aliquots
of LC21 and ODP 967 for BHT isomer analysis. Samples were acetylated in 0.5 mL of
a 1:1 (v:v) mixture of pyridine and acetic anhydride at 50 °C for 1 h, then overnight at
room temperature. Solvent was dried on a 50°C heating block under a stream of $N_2$.
Samples were dissolved in MeOH:propan-2-ol (3:2; v:v), and filtered on 0.2 μm PTFE
filters.
BHT isomer was analysed by high performance liquid chromatography coupled to
positive ion atmospheric pressure chemical ionization mass spectrometry



(HPLC/APCI-MS), using a data-dependent (3 events) scan mode on a system
equipped with an ion trap MS (Talbot et al., 2007;van Winden et al., 2012). Semi-
quantification of BHT isomer was achieved at Newcastle University using a BHT
standard gifted by M. Rohmer.
2.5.2. Analysis of non-derivatised BHT isomer (NIOZ)
BHT isomer of the high resolution S5 sapropel (64PE406-E1) was measured on non-
derivatised aliquots of BDEs using an ultra high performance liquid chromatography
(UHPLC)-Q Exactive Orbitrap MS with electrospray ionisation (Thermo Fischer
Scientific, Waltham, MA), using a method for analysis of intact polar lipids according
to (Wormer et al., 2013). Briefly, separation was achieved on an Acquity BEH C18
column (Waters, 2.1x150 mm, 1.7μm) maintained at 30°C, using (A)
MeOH/$H_2$O/formic acid/14.8 M $NH_{3aq}$ (85:15:0.12:0.04 [v/v/v/v]) and (B)
IPA/MeOH/formic acid/14.8 M $NH_{3aq}$ (50:50:0.12:0.04 [v/v/v/v]) as eluent. The elution
program was: 95% A for 3 min, a linear gradient to 40% A at 12 min, and then to 0%
A at 50 min, which was maintained until 80 min. The flow rate was 0.2 mL $min^{-1}$.
Positive ion ESI settings were: capillary temperature, 300°C; sheath gas ($N_2$) pressure,
40 arbitrary units (AU); auxiliary gas ($N_2$) pressure, 10 AU; spray voltage, 4.5 kV; probe
heater temperature, 50°C; S-lens 70 V. Target lipids were analyzed with a mass range
of $m/z$ 350–2000 (resolution 70,000 ppm at $m/z$ 200), followed by data-dependent
tandem $MS^2$ with parameters as described by Besseling et al., (2018). The combined
extracted ion currents (within 3 ppm) of the protonated, ammoniated, and sodiated
adducts ($m/z$ 547.47209 + 564.49864 + 569.45403, respectively) were used to
integrate BHT isomer. The relative abundance of peak area does not necessarily
reflect the actual relative abundance of the different compounds; however, this method
allows for comparison between the samples analyzed in this study. BHT and BHT
isomer were baseline separated, and the $MS^2$ spectra of BHT and its isomer (Fig. S1)
were comparable to spectra of non-derivatised BHT published by Talbot et al. (2016b).
MS performance was continuously monitored and matrix effects were assessed using
the PAF standard. Peak areas were corrected accordingly. However, as no
commercially available authentic standards were available for non-derivatised BHPs,
semi-quantitative BHT isomer abundance is reported as the integrated peak area
response (response unit, r.u.) for the Levantine S5 (64PE406-E1) record. Although
quantification in not possible, this method does allow for comparison of BHT isomer



abundances between samples as response factors should be identical across the S5
sample set.

### 2.5.3.  Analysis of ladderane fatty acids

Methylated fatty acid fractions were dissolved in acetone, filtered through 0.45 μm, 4
mm diameter PTFE filters, and analysed by high performance liquid chromatography
coupled to positive ion atmospheric pressure chemical ionization tandem mass
spectrometry (HPLC/APCI-MS/MS) in selective reaction monitoring mode to detected
the four ladderane fatty acids and two short-chain ladderane fatty acids (Hopmans et
al., 2006; modified by Rush et al., 2011). Ladderanes were quantified using external
calibration curves of three standards of isolated methylated ladderane fatty acids ($C_{14}$-
[3]-ladderane fatty acid, $C_{20}$-[3]-ladderane fatty acid, and $C_{20}$-[5]-ladderane fatty acid)
(Hopmans et al., 2006;Rush et al., 2011;Rattray et al., 2008).



3.  Results and Discussion
To test the hypotheses that (1) anaerobic ammonium oxidation occurred in the water
column during Mediterranean sapropel events, and (2) BHT isomer could be used as
a biomarker for anammox during these events, a suite of Quaternary and Pliocene
sapropels were examined.

269         3.1. Anammox lipids in S1 – S5 sapropels from the Aegean Sea

Sapropels spanning four of the most recent five events in the Aegean Sea were
sampled from core LC21 from the Aegean Sea and analysed for anammox biomarkers
(Fig. 3a). Ladderane fatty acids (i.e. $C_{18}$-[3]-ladderane fatty acid, and $C_{18}$-[5]-ladderane
fatty acid, $C_{20}$-[3]-ladderane fatty acid, and $C_{20}$-[5]-ladderane fatty acid; Fig. 1), the
traditional biomarkers for anammox bacteria (Jaeschke et al., 2009;Rush et al.,
2012a;Sinninghe Damsté et al., 2002), were found in the most recent sapropel (290 –
610 ng/g TOC; in S1, ~7 ka; Fig. 3a) in abundances comparable to those found in
sediments of the Peru Margin and Arabian Sea (Rush et al., 2012a). Conversely,
ladderanes were not detected in the sediment sampled directly below this sapropel
layer (out S1, Fig. 3a), indicating anammox was an important process during S1
deposition, but likely not before the onset of sapropel deposition. Ladderane
concentration progressively decreased with increasing age of the deeper sapropels:
80 – 170 ng/g TOC in S3 (~85 ka); not detected in S4 (~100 ka); and 0 – 90 ng/g TOC
in S5 (~125 ka). It is worth noting that 2 of the 3 sediments from within S5 did not
contain detectable ladderanes. This demonstrates the previously described sensitivity
of ladderane lipids to diagenesis (Rush et al., 2012a;Jaeschke et al., 2008), and
highlights their potential weakness as a biomarker proxy for past anammox bacteria
in ancient sediments. Residues of TLEs were also saponified for ladderane analysis,
as these have previously been shown to extend the detection of anammox in older
sediments by releasing more matrix-bound ladderanes (Rush et al., 2012a). However,
this did not show any difference in the presence of anammox (i.e. there was no
detection of ladderanes in residues in which the original TLEs did not contain these
biomarkers). The non-detection of ladderanes in most of the S5 samples is particularly
surprising as this is the most intense of the Late Quaternary sapropels (Struck et al.,
2001), having been described as analogous to the modern day Black Sea (Menzel et
al., 2006). Since anammox is currently present and actively removing N in the





redoxcline of the Black Sea (Jensen et al., 2008;Kuypers et al., 2003), it was expected that anammox behaved similarly in the nitrogen cycle of the Eastern Mediterranean during deposition of the S5 sapropel. Given that the oldest detection of ladderanes comes from a slightly older record in the Arabian Sea (Jaeschke et al., 2009), it is unclear why ladderane detection in S5 is sporadic. Perhaps degradation is responsible for the rapid removal of ladderanes from the system during deposition, or the low resolution in the S5 record made these specific sediment depths not ideal targets for anammox activity.

Bacteriohopanetetrol isomer (BHT isomer; Fig. 1) has recently been proposed to be an alternative biomarker for anammox bacteria in paleo-records (Rush et al., 2014b). Our analysis of non-derivatised BHT isomer was based on the previously published method analysing intact polar lipids via reverse phase liquid chromatography (Wormer et al., 2013), and achieved better separation of BHT isomer from BHT compared to the acetylated LC-MS method (cf. Rush et al., 2014b; Fig. S1).The concentration of BHT isomer in the Aegean Sea sapropels showed a similar trend as ladderanes in the shallow sediment layers (Fig. 3b): the concentration was high in S1 (71 – 360 µg/g TOC), and low in the underlying sediment (12 µg/g TOC; out S1), in good agreement with the ladderane data. In contrast, however, BHT isomer was detected in all deeper sapropels at higher concentrations (64 – 180 µg/g TOC in S3; 67 – 90 µg/g TOC in S4; and 68 – 160 µg/g TOC in S5) than the ladderanes. Sediments from outside the sapropel had relatively low, but measurable BHT isomer concentration (8 – 17 µg/g TOC). As BHT isomer was detected in all sapropels, including the oldest S5 sediments, it appears that the rapid removal of ladderanes from the system is due to degradation during deposition. These results clearly demonstrate the utility of BHT isomer as a biomarker for anammox in paleorecords compared to the more labile ladderane lipids. A hemipelagic, light, non-sapropel sediment sampled between S3 and S4 contained neither ladderanes nor BHT isomer (Fig. 3), indicating a period where anammox was likely not active in the Mediterranean nitrogen cycle. Furthermore, the detection of BHT isomer in the non-sapropel sediments underlying S1 and S5 and overlying S3 shows that this lipid is a better biomarker than ladderanes for recording trace amounts of anammox throughout the history of the Mediterranean system, especially in sediment deposited under oxic (bottom) water conditions.



### 3.2. High-resolution evidence shows anammox responds to marine redox shifts in S5 sapropel record

To further investigate the occurrence of anammox during sapropel deposition, we analysed in high resolution the well-developed S5 (TOC content up to 12%; Fig. 4) recovered from the Levantine Basin in the Eastern Mediterranean during a cruise of the R/V Pelagia in 2016 (64PE406-E1; Fig. 2). It was expected that ladderane fatty acids would be preserved in the high TOC sediments of S5. However, in line with the earlier results of ladderane analyses for S5 in the Aegean Sea record, the results from the Levantine Basin were inconclusive. Ladderanes were detected in all, except two, of the thirty sapropel samples, but were at the detection limit (i.e. peak area of 3x background), preventing interpretation of the ladderane profile in S5. The cause of low ladderane concentration even in sediments with high TOC may be due to an unknown degradation mechanism in Mediterranean sapropel sediments.

The BHT isomer does not appears to have been affected by degradation in the same way as ladderane lipids; it was above detection limit in all S5 sediments (Fig. 4b). The concentration of BHT isomer increased progressively by a factor of 10 from the onset of S5 until the core of the sapropel event (from average pre-sapropel value 2.69 E+11 r.u./g TOC to 2.28 E+12 r.u./g TOC at 33 – 34 cm core depth; Fig. 4) and then waned until the termination. This indicates that anammox was an important process during the formation of S5, actively removing nitrogen from the marine system. Photic zone euxinia has been observed in cores from the western part of the Eastern Basin during S5 by the identification of isorenieratene (Marino et al., 2007;Rohling et al., 2006). Isorenieratene is a biomarker lipid for photosynthetic, green sulfur bacteria (*Chlorobiaceae*) that require the unique conditions of light, albeit in relatively low abundance, *and* euxinic waters (Overmann et al., 1992). Although anammox bacteria are inhibited by the presence of free sulfide, they likely thrived at the chemocline during deposition of S5. This is the case, for instance, in the modern Black Sea: at 90 m water depth, where oxygen and sulfide concentrations are both low and nitrite and ammonium are readily available, the presence and activity of anammox has been confirmed via rate measurements and ladderane biomarker observations (Kuypers et al., 2003;Jensen et al., 2008).



There are two considerable peaks in BHT isomer that fall outside of the S5 trend (Fig. 4b), occurring at the onset (2.43 E+12 r.u./g TOC; 46 – 47 cm core depth) and termination (1.12 E+12 r.u./g TOC; 16 – 17 cm core depth) of the sapropel. Sea-level rise and freshening of the Mediterranean is believed to have caused a stepwise removal of oxygen and subsequent slow build-up of anoxia ca. 3 kyr before the (massive) freshwater discharge from the African continent instigated the real onset of S5 (Schmiedl et al., 2003). The intense anammox peak pre-sapropel formation could be a response to this marine redox shift. Anammox would have thrived, consuming the residual low-levels of ammonium and nitrite in an anoxic Mediterranean water column. Then, once monsoonal discharge brought in the initial pulse of nutrients from the Nile, the slow-growing anammox bacterial population would have been rapidly outcompeted by heterotrophic denitrifiers consuming sinking organic carbon being produced in the overlying oxic waters. As S5 progressed and N supply became scarcer, anammox would have repopulated the niche of redoxcline N-remover at core sapropel conditions. The peak of BHT isomer observed at S5 termination (Fig. 4) shows that the conditions were again favourable for anammox to thrive. However, this may have occurred at the anoxic sediment-water interface, rather than in the water column, where low concentrations of nitrite and ammonium could have persisted from the degradation of organic matter settling on the seafloor after the re-oxidation of the water column.

Short-chain (SC) ladderane fatty acids (i.e. $C_{14}$-[3]-ladderane fatty acid and $C_{14}$-[5]-ladderane fatty acid; Fig. 1) are oxic biodegradation products of ladderane fatty acids (Rush et al., 2011), and are used to infer exposure of ladderane lipids to oxic conditions either pre- or post-deposition. SC ladderane fatty acids were only detected in three of the S5 sediments, specifically at sapropel onset (46 – 47 cm core depth) and termination (15 – 16 cm and 16 – 17 cm core depth). This implies that during sapropel maximum, anammox was thriving at the Mediterranean chemocline. Anammox detritus would then have sunk through an anoxic (euxinic) 'Black Sea' water column, unexposed to oxygen and the effects of β-oxidation that produces SC ladderane fatty acids (Rush et al., 2011). This has been seen in the modern Cariaco Basin, where ladderanes are observed, but SC ladderanes are absent (Rush et al., 2012a). The presence of SC ladderanes at the onset and termination, yet absence in the core S5 record, could also corroborate the concept of "split-anoxia" (as proposed





for S1 by Bianchi et al., 2006), which hypothesizes for the first 100 to 1000+ years of
sapropel formation euxinia was present as a mid-depth "oxygen minimum zone",
rather than a continuation from the seafloor. During these periods where the water
column was not fully euxinic, ladderanes would have been oxidised to SC ladderanes
in the underlying waters, which would have contained a certain amount of available
oxygen. Alternatively, as productivity waned, sedimentation rates would have
decreased in the Levantine Basin. Lower sedimentation rates at the onset and
termination of S5 would suggest a longer residence time of ladderanes in sediment
that would periodically be exposed to (sub)oxic bottom water conditions. Oxic water
in-flow of pore waters would have stimulated the β-oxidation responsible for SC
ladderane formation (Rush et al., 2011). It is worth noting that in the low-resolution
Aegean Sea sample set (LC21), all sample from S1 and S3 that contained ladderanes
also contained a high concentration of SC-ladderane fatty acids, whereas the singular
S5 sediment did not contain SC ladderanes. This would appear to indicate that the
Aegean water column during S1 and S3 deposition was not fully euxinic, and that S5
in the Aegean mirrored the euxinic Levantine Basin.
Nitrogen isotope ratios ($\delta^{15}$N) values of bulk nitrogen in S5 sediment show a strong
shift towards low values within the sapropel (Fig. 4a), a feature seen in most sapropels
(Calvert et al., 1992;Sachs and Repeta, 1999;Struck et al., 2001;Higgins et al.,
2010;Mobius et al., 2010). This could potentially be explained by either enhanced
diazotrophic $N_2$-fixation because N was limited in the system (Mobius et al., 2010), or
the preferential uptake and burial of $^{14}$N when nitrate is present in excess and primary
producers have the opportunity to fractionate maximally (Calvert et al., 1992). As a
biomarker for N removal from the system was not available, previous work has only
been able to approach this conundrum with evidence for N fixation processes. Using
isotopic evidence of diazotrophic phytoplankton, Sachs and Repeta (1999) and
Higgins et al. (2010) argue that Mediterranean surface water was nitrogen-limited
during sapropel events. Here, for the first time, we present evidence of N loss in a
Mediterranean sapropel using BHT isomer as an anammox biomarker. The fact that
BHT isomer concentration increases towards the core of S5 may appear to suggest
that N species were not limited, and rather that freshwater run-off could be resupplying
these nutrients to microorganisms in the water column and enhancing the pool of N.
However, anammox thrive at the redoxclines of modern oxygen minimum zones



(Pitcher et al., 2011;Rush et al., 2012b) and euxinic basins (Wakeham et al.,
2012;Kuypers et al., 2003), where pulses of "fresh" N species do not necessarily
reach. At the S5 'Black Sea type' redoxcline, anammox did not need a riverine supply
of N, but could have instead been sustained by the advection of N from deeper waters
(Rohling et al., 2006) or by N remineralised from the sinking pool of (diazotrophic)
organic matter from above. We can interpret BHT isomer results as N removal by
anammox was at its highest flux during core S5 sapropel conditions, and that the
anammox process appears to play an integral role in N cycling during sapropel events.
3.3. Anammox distribution varies between sapropel formations: evidence from a
434        Pliocene sapropel event

To confirm that anaerobic ammonium oxidation has occurred throughout the history of
anoxia in the Mediterranean basin, not only in the most recent Quaternary sapropels,
BHT isomer concentration was analysed across a high-resolution Pliocene sapropel
(ODP Leg 160, Site 967; Fig. 2). The anammox biomarker is present throughout this
older record (Fig. 5b). Yet here, BHT isomer displayed a distribution different to that
of the S5 record. Sapropel S73 showed two distinct peaks in BHT isomer: at the onset
(110 – 240 µg/ g TOC; 69 – 73 cm core depth) and at the termination (640 – 1100 µg/g
TOC; 54 – 59 cm core depth) of the sapropel, much like the trend seen in the S5
Levantine sapropel. However, BHT isomer concentration was low during the core
Pliocene sapropel event (Fig. 5b), likely representing unfavourable conditions for
anammox during this sapropel. Isorenieratene has been detected in the Pliocene
record of Site 967, albeit in a different sapropel event (Menzel et al., 2002). It is
possible that euxinia shoaled further into the photic zone during this Pliocene sapropel,
forcing anammox at the chemocline to compete for N with phytoplankton. Anammox
would have therefore only thrived during the build-up and termination periods when
photic zone euxinia would have been deeper/less intense. There was a spike in BHT
isomer concentration mid-sapropel that coincided with a decrease in TOC (65 – 67 cm
core depth; Fig. 5a). Mid-sapropel breaks have been reported elsewhere, as
repopulation events of benthic fauna (e.g. Rohling et al., 1993), and could be due to
inflow of freshly ventilated deep-water. The concentration of BHT isomer was still high
after sapropel deposition (~250 µg/g TOC; <40 cm core depth), relative to that pre-
sapropel. This may indicate that the anammox process remained an important N
process in the Mediterranean after bottom water anoxia waned.





Combined, the high-resolution results from the S5 and Pliocene sapropels indicate that the functioning of anammox is not always the same during periods of Mediterranean anoxia. This demonstrates that the response of the N cycle to anoxic conditions can vary drastically from one sapropel event to the next.

## 4. Conclusion

BHT isomer, a lipid synthesised by marine anaerobic ammonium oxidising (anammox) bacteria, was detected at high concentration in all Mediterranean sapropel sediments. This study highlights the potential of BHT isomer as a biomarker for anammox during past periods of basin-wide anoxia. It is also apparent that the response of anammox to shifts in redox conditions during anoxia is not consistent between sapropel events. The anammox peak in S5 occurred during core sapropel conditions, whereas anammox responded in an opposite trend in the Pliocene sapropel record.

Investigating the variability of anammox in these sapropel events may enhance our understanding of N cycling during other periods of intense organic matter deposition in the past. Sapropel features have been found in the sediment records of different marginal seas (e.g. Japan Sea, Red Sea; cf. Emeis et al., 1996). The restricted paleogeography during Oceanic Anoxic Events is also thought to have contributed to the propagation of anoxia in the Cretaceous and Jurassic. BHT isomer can possibly be used to explore the role anammox may have played in these basin anoxic events. The residence time of BHT isomer in marine sediment records likely does not extend beyond the Early Cretaceous (van Dongen et al., 2006;Talbot et al., 2016a). However, BHT isomer can be applied to the Paleocene-Eocene Thermal Maximum (PETM; 55 Ma). Thermally stable lipid products of anammox biomass (Rush et al., 2014a) could serve as alternative biomarkers for anammox in more mature sediments from the Cretaceous and Jurassic. Furthermore, investigating the compound-specific isotope values of BHT isomer in a marine sample set will strengthen the use of BHT isomer as a biomarker for anammox.



## 5. Acknowledgement

Guy Rothwell is thanked for his help collecting LC21 samples from the BOSCORF repository. We thank the Captain and crew of the R/V Pelagia for the collection of the sapropel S5 (cruise 64PE406). Pieter Dirksen subsampled the S5 record. Denise van der Slikke-Dorhout and Çağlar Yildiz are acknowledged for extracting the S5 record. We are grateful to the Ocean Drilling Program (ODP) for the samples used in this study as well as to the ODP Core Repository (Bremen, Germany) where Luke Handley and Thomas Wagner were involved in the collection of ODP Leg 160 sapropel sequences. Eelco Rohling is thanked for initial discussion about sapropel sampling. Michel Rohmer is thanked for gifting the original BHT standard to Helen Talbot at Newcastle. This work was supported by the Natural Environment Research Council (NERC) project ANAMMARKS (NE/N011112/1) awarded to DR. This work was also supported by funding from the Netherlands Earth System Science Center (NESSC) though a gravitation grant (NWO 024.002.001) from the Dutch Ministry for Education, Culture and Science to JSSD. NERC (Grant number NE/E017088/1) and the European Research Council (ERC) (Starting Grant No. 258734 awarded to HMT for project AMOPROX) are gratefully acknowledged for partially funding this research.





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



Figure Captions

Figure 1. Structures of anammox biomarker lipids used in this study. Bacteriohopanetetrol (BHT); bacteriohopanetetrol stereoisomer (BHT isomer), unknown stereochemistry; ladderane fatty acids with 3 or 5 cyclobutane moieties and 18 or 20 carbon atoms; short-chain ladderane fatty acids with 3 or 5 cyclobutane moieties and 14 carbon atoms.

Figure 2. Map of the eastern Mediterranean showing the locations of sapropel cores used in this study. LC21: low-resolution S1, S2, S3, and S5 sapropels from the Aegean Sea; 64PE406: high-resolution S5 sapropel from the Levantine Basin; ODP 967: high-resolution Pliocene sapropel from the Levantine Basin. Map created with SimpleMappr: Shorthouse, David P. 2010. SimpleMappr, an online tool to produce publication-quality point maps.

Figure 3. Box plots of (a) ladderane fatty acid concentration and (b) BHT isomer concentration in four recent sapropels (S1 - S5; 7 - 125 ka) from the Aegean Sea (R/V Marion Dufresne LC21). Lines are the mean markers and circles denote data points.

Figure 4. (a) Total organic carbon (TOC) content, isotope values of bulk nitrogen ($\delta^{15}N$) and carbon ($\delta^{13}C$), and (b) BHT isomer concentration through a high resolution S5 sapropel record from site 64PE406 (R/V Pelagia) in the Levantine Basin. Core photo provided by R. Hennekam.

Figure 5. Total organic carbon (TOC) content (a) and BHT isomer concentration (b) through a Pliocene sapropel (2.97 Ma) from the Levantine Basin (ODP Leg 160 Site 967). Core photo provided by L. Handley.






Figures
Figure 1

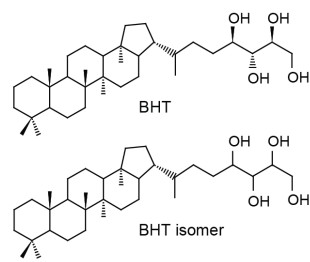


Figure 2

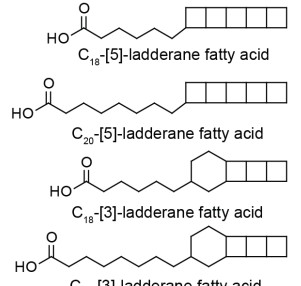




Figure 3

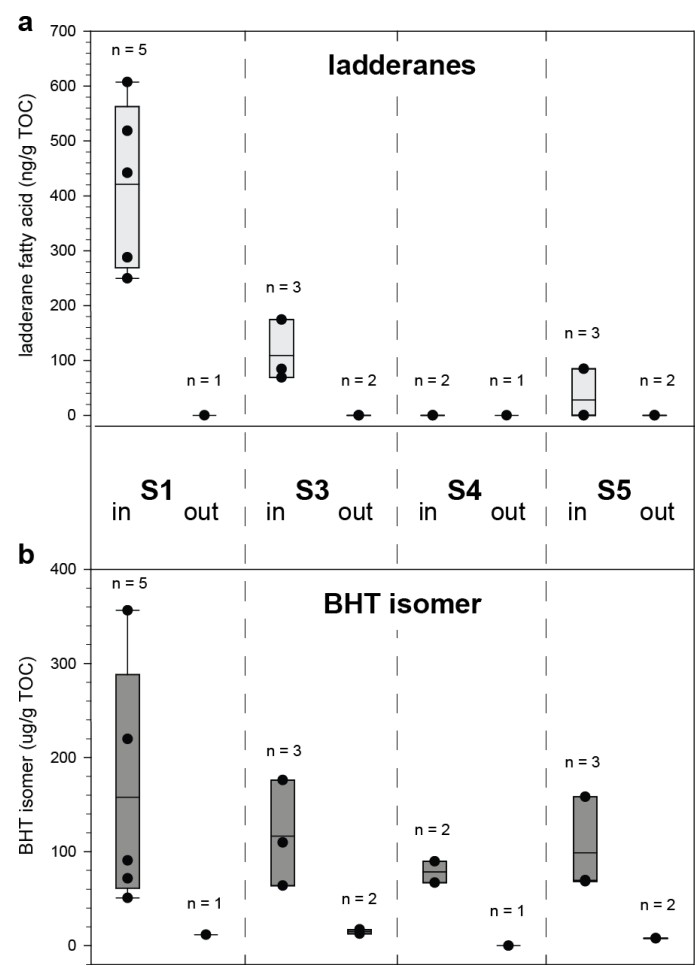











Figure 4

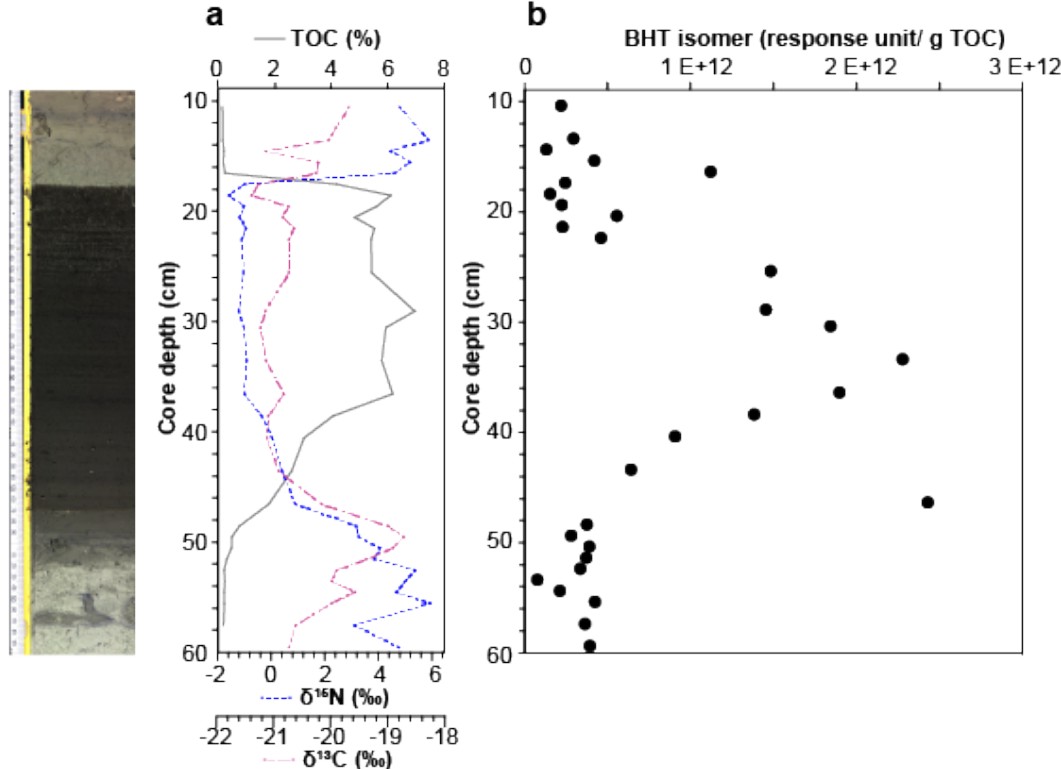









Figure 5

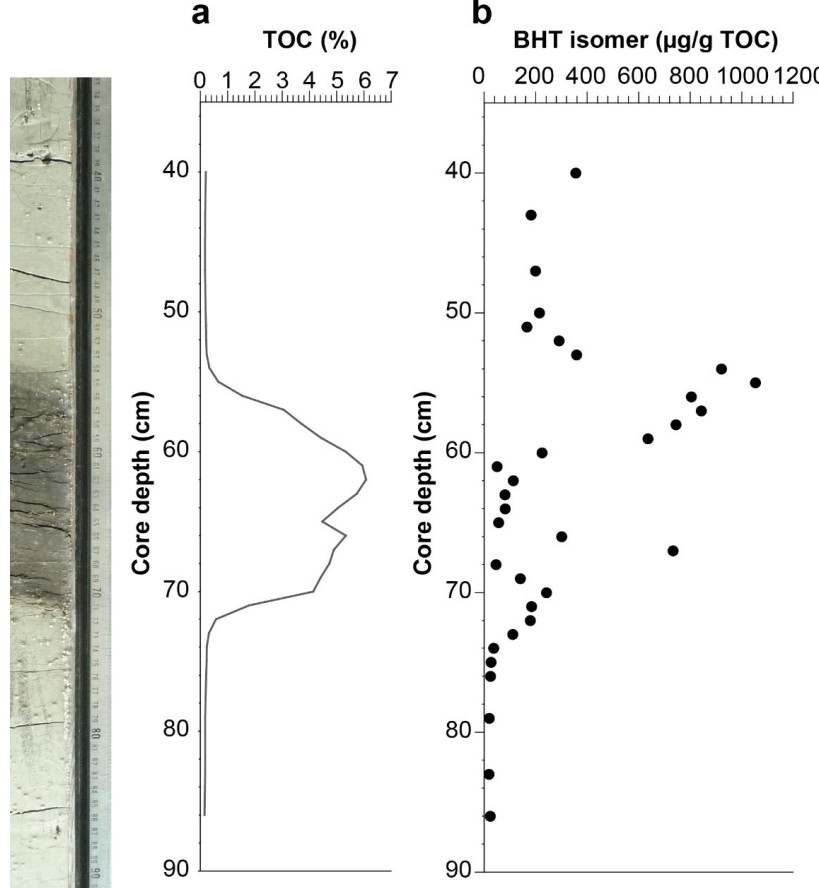







