# Peer review of "Biomarker evidence for the occurrence of anaerobic ammonium oxidation in the eastern Mediterranean Sea during Quaternary and Pliocene sapropel formation"

_Biogeosciences, 2019_

## Referee Comment (RC1) · Anonymous Referee #1 · 1 Mar 2019

This manuscript is a nice contribution to Biogeosciences and provides valuable new data exploring the applicability of an up and coming biomarker for anammox. The data from the Pliocene S73 sapropel are particularly interesting since they extent the timescale on which BHT isomer has been used as a proxy for anammox. Overall, I have relatively little criticism. Please find some open questions and points that need clarification below.

General comments 1) While I agree that Fig. 3 provides a good argument for an anammox origin of the BHT isomer in the investigated Aegean sapropels, this compound

has been identified in various non-anammox settings/under different redox conditions (the authors also point this out). So far, the most convincing argument for an anammox origin of the BHT isomer in environmental samples seems to come from BHT isomer/(BHT+BHT isomer) ratios, which are really high in anammox cultures (Rush et al. 2014) and also in OMZ settings (Matys et al. 2017). In this respect, I would strongly encourage the authors to include the BHT data as well, to support the argument that the trends seen in BHT isomer abundances across sapropel horizons reflect the occurrence/increase of anammox during sapropel deposition. The sapropels have very high TOC in comparison to the under- and overlying sediments. Thus, these sediments can be expected to be hot spots for deep biosphere bacterial communities, which could also be (non-anammox) producers of BHT isomer and increasing absolute BHT isomer abundances for example could simply reflect a relative increase of bacterial over eukaryotic biomass (or some other process). Can this be excluded? I think it would helpful to include the BHT data and to show BHT isomer/(BHT+BHT isomer) ratios for these records (Figs. 4 and 5) to strengthen the argument for an anammox biomarker. If BHT isomer/(BHT+BHT isomer) ratios show different or less obvious trends, please add a respective section/paragraph to the discussion.

2) The occurrence of SC ladderanes and simultaneous absence of ladderanes (or occurrence only at the detection limit) in the S5 sapropel in core 64PE406 warrants some more discussion. While SC ladderanes could only be detected in 3 samples, the authors provide two possible $\beta$-oxidation scenarios to explain their occurrence. However, what is missing is the explanation why there are no ladderanes in the maximum sapropel unit for which fully anoxic (euxinic) conditions and peak anammox are invoked. I would expect that the preservation potential for ladderanes was much higher during maximum sapropel deposition. If this was only due to an "unknown" degradation mechanism (as stated earlier), shouldn't i) that mechanism also degrade the SC ladderanes generated during the onset and termination of the sapropel? or ii) if that "unknown" mechanism abruptly starts/ends during the onset/termination of sapropel deposition, what kind of mechanism would work opposite to the "normal" redox-driven preservation/degradation mechanisms, i.e., higher degradation under anoxic conditions? Based on the BHT isomer abundances, the ladderane pattern does not seem to be driven by productivity since the onset and termination BHT isomer peaks are not significantly higher than the maximum sapropel concentrations and a productivity argument would also disagree with the peak anammox assumption during maximum sapropel deposition made earlier. Please elaborate. Also, please consider including the SC ladderane abundances on a second axis in the b panel of Fig. 4, it will help guiding the reader through the arguments.

Specific comments l. 140: change to "immediately" l. 163: change to "ratio" l. 257: change to "detect" l. 324-327: since the BHT isomer has also been detected in non-anammox samples, the trace amounts detected in the background non-sapropel samples may also reflect different bacterial sources rather than minimal anammox activity. Again, BHT isomer/(BHT+BHT isomer) ratios would help here. l. 339-340: please elaborate a little more which kind of (unknown) mechanisms you consider may cause ladderane decomposition (see also above comment 2). l. 341: change to "appear" l. 349: change to "S5 formation" l. 403: change to "all samples" l. 439-443: the argument is contradictory, in the first sentence it says that the "BHT isomer displayed a distribution different to that of the S5 record" while the second sentence states "much like the trend seen in the S5 Levantine sapropel." Please clarify. l. 446-448: "It is possible that euxinia shoaled further into the photic zone during this Pliocene sapropel, forcing anammox at the chemocline to compete for N with phytoplankton." If the euxinia was even more pronounced, wouldn't one expect to find isorenieratene (or other biomarkers such as okenone/okenane) in sapropel 73 if it was found in a different sapropel at this site? Please elaborate a little. l. 450-452: "There was a spike in BHT isomer concentration mid-sapropel that coincided with a decrease in TOC (65 – 67 cm core depth; Fig. 5a)." To me it appears that the BHT isomer spike pre-dates the TOC decrease, which is similar to the pattern observed for the S5 in core 64PE406 but opposite to the pattern evident for the termination of the S73 when TOC decreases earlier. How is this explained?

Fig. 3 Please only show data points, not box plots. The minimum sample size for box plots is n=5, which would allow only the S1 data to be visualized this way. See further Krzywinski M. and Altman N. (2014) Visualizing samples with box plots. Nature Reviews Microbiology 11, 119–120. However, why not plot the Aegean core like the other two cores (Y-axis could be broken between sapropels)? This would allow better comparison with the other records as well.

Fig. 4 Maybe the quality of this figure could be improved, Fig. 5 has much better resolution. For both figures, a thin line connecting the circles in panel b would aid at seeing the trends.

Supplement The supplementary figure (showing the chromatograms and the mass spectrum of non-acetylated BHT and BHT isomer) is not referred to in the text and it does not have a caption.

---

## Referee Comment (RC2) · Cecile Blanchet (Referee) · 14 Mar 2019

This new paper explores the occurrence of anammox (anaerobic oxidation of ammonium) in the water column of the Eastern Mediterranean during sapropel deposition. Not being a biomarker or anammox specialist but interested in chemical processes in low-oxygen environments, I found this publication very insightful and well presented. I have only minor comments that aim at clarifying the message.

General comments:

1) It might be useful for non-anammox specialists to draw a little sketch to show where you expect anammox to take place in the water column (e.g. from present-day OMZ) and its relationships with euxinia and anoxia (for instance with schematic O2 and H2S profiles, chemocline, redoxcline...) and competition with phytoplankton. It would also help to visualize the interpretations that you discuss regarding the patterns of anammox in the various sapropels.

2) In general, I am missing a bit a comparison between the interpretations drawn here in term of water-column chemistry with other type of data. For instance, relationships between the build-up of anoxia or the presence of euxinia are mentioned in the text but do not appear in the figures. For S5 at higher resolution (64PE406), it might be useful to give temporal indications so that it can be compared to other records. Along a similar line, the relationships between deep-water stagnation, eutrophication and eutrophia have been widely explored for S5 and it might be useful to place your record in a wider context (also to highlight its relevance).

3) Another point which I am missing is a more structured discussion on the effects of post-depositional diagenesis on your markers. Diagenesis associated with changes in sedimentation rates and level of TOC in and around sapropels has been well-documented and generally allows to identify specific horizons in sapropels layers (proto-sapropel, oxidized "burn-down" sapropels). Higher BHT isomer values and the presence of SC ladderanes below and above S5 and the Pliocene sapropel should be discussed in this context.

4) What can help you decipher whether anammox occurred in the water column or in the sediments? I understand that the presence of free sulfides is preventing anammox to occur but would anammox happen in sediments where the overlying water is not euxinic and where sulfates are present (say until the sulfate-methane transition zone)? This is related to my previous points and questions the role of sediment-bound anammox in your records: would processes occurring during early diagenesis (i.e., when redox and chemical fronts shifted) in the sediments be able to trigger anammox (and

overprint the water-column derived biomarker record)? Is it possible for anammox to occur in the sediment core after retrieval and during storage? This might help understand why there are ladderanes in S5 in LC21 but not in 64PE406: i) storage and sediment handling artefact, ii) "unknown degradation mechanism" or iii) spatially non-uniform occurrence of anammox (e.g., in the Aegean but not in the Levantine Basin)?

5) Finally, can you rule out that anammox biomarkers were not brought to the core site by runoff (say a "detrital/exogenous" anammox component)? If I am not mistaken, anammox occurs in freshwater and coastal environments as well, but would the BHT isomer biomarker resist fluvial transportation and exposition to oxic conditions?

Specific comments:

I agree with reviewer #1 that information is missing in the figures:

Fig. 3: add data for BHT isomers in other cores (S5 for 64PE406 and S73 for ODP 160)

Fig. 4: it would indeed be insightful to show ratios and SC ladderanes (see comments by reviewer #1). Drawing a line between points would also be helpful. The depth scale can be removed for the plot 4b (and generally, a and b are not needed). If possible, indicate the various sub-layers in the sapropel (proto-sapropel, oxidized sapropel) using the Ba and Mn concentrations (or as ratio over Al or Ti). Ba is a good indicator for sapropel extend and Mn shows the upper extend (upper redox front), so the oxidized part of the sapropel (where the TOC is low). If you have some time indication, it might be interesting to indicate/plot some results from other records (isorenioratene, forams, etc...) to get a fuller picture of the changes in water-column properties.

Such a figure (depth profile) is missing for LC21, although a lot of data has been gathered on this core. This would allow direct comparison between other proxies and the anammox biomarkers, even at low sampling resolution.

Fig. 5: please also connect dots with a line in 5b and if possible, indicate the various

horizons in the sapropel (see comments for Fig. 4).

While reading section 3.1, I was wondering why ladderanes had not been measured in 64PE406, and it is only when I read section 3.2 that I got my answer. It should be clear from the beginning that ladderanes were measured both in LC21 and 64PE406 (also in the method part, section 2.4.2) but that they could not be detected in the latter one.

Introduction line 48-54: perhaps introduce the meaning of anoxia vs. euxinia for non-specialists? In general, it would be more accessible if terms would be better introduced (e.g., chemocline vs. redoxcline) or shown on schematic representations.

l. 365-371: I find this part quite obscure: what is meant by "Then, once monsoonal discharge brought in the initial pulse of nutrients from the Nile, [...]"? I do not follow the order of events. Perhaps making that appearing on fig. 4 would be helpful (e.g., by comparing to timing of freshwater pulses and development of anoxia)? Or draw small sketches?

Similarly, with the proposal that the observed signal might be related to "split-anoxia": not very clear why that happens and might be useful to provide a visualization.

But once again, I enjoyed reading this paper and feel that it will contribute value to our understanding of changes in the marine environment related to deoxygenation processes, which were recently highlighted as a growing concern for present oceanic basins.

---

## Author Comment (AC1) · 27 Mar 2019

On behalf of the co-authors, I thank this anonymous referee for their helpful comments and the time they took to improve the paper. I would also like to apologies for the delayed reply to this review, which was caused by fieldwork being carried out in a poor-internet area when the review came.

We respond to the individual comments below (in bold):

*This manuscript is a nice contribution to Biogeosciences and provides valuable new data exploring the applicability of an up and coming biomarker for anammox. The data from the Pliocene S73 sapropel are particularly interesting since they extent the timescale on which BHT isomer has been used as a proxy for anammox. Overall, I have relatively little criticism. Please find some open questions and points that need clarification below.*

*General comments*

*1) While I agree that Fig. 3 provides a good argument for an anammox origin of the BHT isomer in the investigated Aegean sapropels, this compound has been identified in various non-anammox settings/under different redox conditions (the authors also point this out). So far, the most convincing argument for an anammox origin of the BHT isomer in environmental samples seems to come from BHT isomer/(BHT+BHT isomer) ratios, which are really high in anammox cultures (Rush et al. 2014) and also in OMZ settings (Matys et al. 2017). In this respect, I would strongly encourage the authors to include the BHT data as well, to support the argument that the trends seen in BHT isomer abundances across sapropel horizons reflect the occurrence/increase of anammox during sapropel deposition. The sapropels have very high TOC in comparison to the under- and overlying sediments. Thus, these sediments can be expected to be hot spots for deep biosphere bacterial communities, which could also be (non-anammox) producers of BHT isomer and increasing absolute BHT isomer abundances for example could simply reflect a relative increase of bacterial over eukaryotic biomass (or some other process). Can this be excluded? I think it would helpful to include the BHT data and to show BHT isomer/(BHT+BHT isomer) ratios for these records (Figs. 4 and 5) to strengthen the argument for an anammox biomarker. If BHT isomer/(BHT+BHT isomer) ratios show different or less obvious trends, please add a respective section/paragraph to the discussion.*

**We agree with the reviewer that BHT isomer ratio is a useful tool to disentangle the contribution of anammox from other bacterial sources to the BHT pool. However, we originally chose not to include the BHT ratio in Figures 4 and 5 solely because, in general, the ratio follows the same trend as the BHT isomer concentration. This also led us to conclude that the sole source of BHT isomer in all samples was marine anammox. However, we agree that visually it would be helpful to include the proportion of BHT isomer relative to BHT in these two records, and we will amend the figures in the revised manuscript to include BHT isomer ratio.**

*2) The occurrence of SC ladderanes and simultaneous absence of ladderanes (or occurrence only at the detection limit) in the S5 sapropel in core 64PE406 warrants some more discussion. While SC ladderanes could only be detected in 3 samples, the authors provide two possible β-oxidation scenarios to explain their occurrence. However, what is missing is the explanation why there are no ladderanes in the maximum sapropel unit for which fully anoxic (euxinic) conditions and peak anammox are invoked. I would expect that the preservation potential for ladderanes was much higher during maximum sapropel deposition. If this was only due to an "unknown" degradation mechanism (as stated earlier), shouldn't i) that mechanism also degrade the SC ladderanes generated during the onset and termination of the sapropel? or ii) if that "unknown" mechanism abruptly starts/ends during the onset/termination of sapropel deposition, what kind of mechanism*

*would work opposite to the "normal" redox-driven preservation/degradation mechanisms, i.e., higher degradation under anoxic conditions? Based on the BHT isomer abundances, the ladderane pattern does not seem to be driven by productivity since the onset and termination BHT isomer peaks are not significantly higher than the maximum sapropel concentrations and a productivity argument would also disagree with the peak anammox assumption during maximum sapropel deposition made earlier. Please elaborate. Also, please consider including the SC ladderane abundances on a second axis in the b panel of Fig. 4, it will help guiding the reader through the arguments.*

**We were equally surprised by ladderanes being detection-limited, especially within the core of the S5 sapropel, where we also expected the highest preservation potential. As this reviewer points out, ladderane removal appears to be most intense during peak anoxia, where anammox was an important process, as evident by high BHT isomer concentration in the sapropel core. Furthermore, we argue that the two peaks in BHT isomer concentration found at the onset and termination intervals of the sapropel are evidence that anammox thrived above a water column that was not yet fully euxinic. It is in these intervals outside the core sapropel where we also find the only three occurrences of short-chain (SC) ladderanes. SC ladderanes are the result of oxic β-oxidation of ladderanes (Rush et al., 2011). Detrital anammox at the onset and termination of the sapropel would have been exposed to low levels of oxygen as it sunk through the water column. These waning sapropel intervals are the only time oxygen was present for this β-oxidation to occur. However, the unknown mechanism that we postulate removed ladderanes during the core sapropel would have had to be active under anoxic (euxinic) conditions. However, as anaerobic degradation experiments on anammox biomass have not been performed, we cannot suggest what kind of ladderane degradation reaction occurs under anoxic conditions, nor what the resulting diagenetic product(s) might be. Future work should include anoxic degradation experiments on anammox biomass to elucidate potential mechanisms.**

**Thus, we speculate that the original ladderanes present in the sapropel water column, which we do not find back in the sapropel sediment archive, were removed either by i) the unknown process under anoxic conditions in the core sapropel or ii) the β-oxidation process under oxic conditions at sapropel onset and termination. However, the result is the same: both processes brought about the removal of the original ladderane fatty acids. We will amend Figure 4 to indicate at which depth intervals we find SC ladderanes to clarify our discussion.**

*Specific comments l. 140: change to "immediately" l. 163: change to "ratio" l. 257: change to "detect" l. 324-327: since the BHT isomer has also been detected in nonanammox samples, the trace amounts detected in the background non-sapropel samples may also reflect different bacterial sources rather than minimal anammox activity. Again, BHT isomer/(BHT+BHT isomer) ratios would help here. l. 339-340: please elaborate a little more which kind of (unknown) mechanisms you consider may cause ladderane decomposition (see also above comment 2). l. 341: change to "appear" l. 349: change to "S5 formation" l. 403: change to "all samples" l. 439-443: the argument is contradictory, in the first sentence it says that the "BHT isomer displayed a distribution different to that of the S5 record" while the second sentence states "much like the trend seen in the S5 Levantine sapropel." Please clarify. l. 446-448: "It is possible that euxinia shoaled further into the photic zone during this Pliocene sapropel, forcing anammox at the chemocline to compete for N with phytoplankton." If the euxinia was even more pronounced, wouldn't one expect to find isorenieratene (or other biomarkers such as okenone/okenane) in sapropel 73 if it was found in a different sapropel at this site? Please elaborate a little.*

**We will amend the wording of the revised manuscript to reflect the changes suggested above. To clarify the discussion about the position of anammox in the water column during S73: we agree**

with the reviewer that based on our hypothesis that the anammox position shoaled, we could expect molecular evidence of photic zone euxinia in this S73 sapropel. Unfortunately, the analysis of the biomarkers suggested here by the reviewer (i.e. isoreneriantene and okenones) did not fall within the scope of the lab work undertaken for this manuscript. We expect future work on these sapropel samples in the coming years to determine the presence/absence of photic zone euxinia biomarkers.

*l. 450-452: "There was a spike in BHT isomer concentration mid-sapropel that coincided with a decrease in TOC (65 – 67 cm core depth; Fig. 5a)." To me it appears that the BHT isomer spike pre-dates the TOC decrease, which is similar to the pattern observed for the S5 in core 64PE406 but opposite to the pattern evident for the termination of the S73 when TOC decreases earlier. How is this explained?*

We suggest that the peak in BHT isomer at this depth interval in S73 is due to a freshening of the deep waters. However, the reviewer correctly points out that the peak in BHT isomer starts before this event was reflected as a decrease in TOC. It is possible that the removal of euxenic conditions by this reventilation would have directly stimulated anammox bacteria that was inhibited by a build-up of sulfide, but that the impact on TOC would have been slightly delayed. We will amend this section to include this point.

*Fig. 3 Please only show data points, not box plots. The minimum sample size for box plots is n=5, which would allow only the S1 data to be visualized this way. See further Krzywinski M. and Altman N. (2014) Visualizing samples with box plots. Nature Reviews Microbiology 11, 119–120. However, why not plot the Aegean core like the other two cores (Y-axis could be broken between sapropels)? This would allow better comparison with the other records as well.*

In the work up of this manuscript, we attempted to show these data on a broken scale, as suggested by the reviewer, but the resulting figure was unclear in our opinion, due to the small number of samples for each sapropel. However, in order to follow the suggestion of the reviewer, we will remove the box plots and only show the scattered data for this figure.

*Fig. 4 Maybe the quality of this figure could be improved. Fig. 5 has much better resolution. For both figures, a thin line connecting the circles in panel b would aid at seeing the trends.*

We agree with the reviewer that the quality of Figure 4 compared to Figure 5 does appear worse in the manuscript. However both figures were generated in the same software, and we have no explanation for why their qualities differ. We will monitor them in the post-production of the revised manuscript to ensure Fig. 4 is high-quality. We will connect data points with a line in the amended figures.

*Supplement The supplementary figure (showing the chromatograms and the mass spectrum of non-acetylated BHT and BHT isomer) is not referred to in the text and it does not have a caption.*

The caption of the supplemental figure was lost in post-production and should read: Supplemental Figure 1. Base peak chromatogram (a) and combined extracted ion 688 currents (within 3 ppm) of protonated, ammoniated, and sodiated adducts (m/z 689 547.47209 + 564.49864 + 569.45403, respectively) (b) of non-derivatised BHT and 690 BHT isomer in sediment from 64PE406-E1 core depth 46 – 47 cm. (c) Combined 691 orbitrap HRMS2 of 6 scan events over the ammoniated adduct ([M+NH4]+; m/z 692 564.49864) of BHT isomer.
The reviewer rightly points out that we did not reference this figure in the manuscript. This will be amended in the revised version.

---

## Author Comment (AC2) · 29 Mar 2019

We thank Dr Blanchet for her thorough and helpful comments which will greatly improve the quality of this manuscript.

**We have replied to the reviewer comments below (in bold):**

*This new paper explores the occurrence of anammox (anaerobic oxidation of ammonium) in the water column of the Eastern Mediterranean during sapropel deposition. Not being a biomarker or anammox specialist but interested in chemical processes in low-oxygen environments, I found this publication very insightful and well presented. I have only minor comments that aim at clarifying the message.*

*General comments:*

*1) It might be useful for non-anammox specialists to draw a little sketch to show where you expect anammox to take place in the water column (e.g. from present-day OMZ) and its relationships with euxinia and anoxia (for instance with schematic O2 and H2S profiles, chemocline, redoxcline. . .) and competition with phytoplankton. It would also help to visualize the interpretations that you discuss regarding the patterns of anammox in the various sapropels.*

**We agree that a cartoon showing the different anammox scenarios in the water column during sapropel events will be useful and we will include such a figure in the revised manuscript.**

*2) In general, I am missing a bit a comparison between the interpretations drawn here in term of water-column chemistry with other type of data. For instance, relationships between the build-up of anoxia or the presence of euxinia are mentioned in the text but do not appear in the figures. For S5 at higher resolution (64PE406), it might be useful to give temporal indications so that it can be compared to other records. Along a similar line, the relationships between deep-water stagnation, eutrophication and eutrophia have been widely explored for S5 and it might be useful to place your record in a wider context (also to highlight its relevance).*

**We agree with Dr Blanchet that creating a graphical representation of the proposed anammox interpretations that are already made in the text is an excellent idea. We will include these (e.g. references to the build-up of anoxia) in the new cartoon figure (mentioned above).**

*3) Another point which I am missing is a more structured discussion on the effects of post-depositional diagenesis on your markers. Diagenesis associated with changes in sedimentation rates and level of TOC in and around sapropels has been welldocumented and generally allows to identify specific horizons in sapropels layers (proto-sapropel, oxidized "burn-down" sapropels). Higher BHT isomer values and the presence of SC ladderanes below and above S5 and the Pliocene sapropel should be discussed in this context.*

**Though not mentioned in the text or figures, the high-resolution S5 sapropel (Fig. 4) did not show any evidence of post-depositional "burn-down" diagenetic alterations. We will incorporate these findings (as a reference to a recently accepted manuscript that describes XRF measurements of this sapropel (Dirksen et al., 2019)) into the discussion of the revised submission. Unfortunately, the same analyses were not performed on the Pliocene sapropel. However, we would not expect that any potential TOC burndown would selectively preserve organic biomarker lipids. It is more likely that if burndown were to have occurred, BHT isomer and ladderanes would also be subjected to these diagenetic processes. Nevertheless, we will include as part of the discussion burndown as a potential factor affecting the Pliocene core.**

*4) What can help you decipher whether anammox occurred in the water column or in the sediments? I understand that the presence of free sulfides is preventing anammox to occur but would anammox happen in sediments where the overlying water is not euxinic and where sulfates are present (say until the sulfate-methane transition zone)? This is related to my previous points and questions the role of sediment-bound anammox in your records: would processes occurring during early diagenesis (i.e., when redox and chemical fronts shifted) in the sediments be able to trigger anammox (and overprint the water-column derived biomarker record)? Is it possible for anammox to occur in the sediment core after retrieval and during storage? This might help understand why there are ladderanes in S5 in LC21 but not in 64PE406: i) storage and sediment handling artefact, ii) "unknown degradation mechanism" or iii) spatially nonuniform occurrence of anammox (e.g., in the Aegean but not in the Levantine Basin)?*

This is an excellent remark. Anammox is known to occur in certain marine sediments (e.g. Trimmer et al. 2003 Appl. Environ. Microbiol.; Jaeschke et al., 2010 Limnol. Oceanogr.). However, anammox activity has only ever been recorded in the upper surface sediment, which is logical as anammox requires both ammonium (which is often available in anoxic sediment) and nitrite (which is rarely detected in sediments deeper than the upper 5-10 cm). Thus, it is very unlikely that anammox would ever been active in sediments at the sulfate methane transition zone. It was also originally suggested that BHT isomer is a biomarker for pelagic anammox (Rush et al., 2014 Geochim. Cosmochim. Acta). However, follow up studies are needed to confirm this.

Furthermore, sedimentary anammox has previously shown a preference for low carbon mineralisation activity, being outcompeted by heterotrophic sedimentary denitrification in sediments with more available reactive carbon (Thamdrup and Dalsgaard, 2002, Appl. Environ. Microbiol.; Engstrom et al. 2005, Geochim. Cosmochim. Acta; Jaeschke et al., 2010 Limnol. Oceanogr.). Therefore, we would expect that if a sedimentary N removal process was to have occurred during the sapropel (or post sapropel deposition) in sediment with high TOC, denitrification would have been favoured over anammox. However, it is more likely that all of the nitrate and nitrite would have been consumed already in the water column of the sapropel, leaving only the accumulation of ammonium in the sediment.

There were ladderanes in the 64PE406 core, but as Dr Blanchet points out in a comment below, this was not clear. We will amend the manuscript to mention these analyses earlier. As discussed in the manuscript, the ladderanes in the 64PE406 core were at detection limit which did not allow for interpretations of the results. As to whether anammox could have been active in stored cores: this is an unlikely scenario in i) the 64PE406 core as samples were immediately frozen after the core was opened and subsampled, and ii) the LC21 core as anaerobic chemolithoautotrophic anammox would not have been encouraged by the presence of oxygen and lack of N substrates in the cold-stored split core. We believe that the storage of the LC21 core would have rather caused the preferential degradation of ladderanes (as these are more labile than BHPs).

*5) Finally, can you rule out that anammox biomarkers were not brought to the core site by runoff (say a "detrital/exogenous" anammox component)? If I am not mistaken, anammox occurs in freshwater and coastal environments as well, but would the BHT isomer biomarker resist fluvial transportation and exposition to oxic conditions?*

We thank the reviewer for bringing this point up. Our response below is also of interest to Reviewer 1's comments about additional bacterial sources of BHT isomer. Anammox is indeed a process that also occurs in freshwater and soil environments. However, only the anammox genus Scalindua is present in marine environments (Villanueva et al., 2014 Front. Microbiol.). Isomers of

**BHT have also been detected in non-anammox bacteria (cf. Rush et al., 2014 Geochim. Cosmochim. Acta). However, we are currently working up a manuscript that shows the BHT isomer synthesized by Scalindua is different from the isomers synthesized by non-marine anammox genera and non-anammox bacteria. Thus, we conclusively identify the BHT isomer present in these samples as an exclusively marine anammox signature.**

*Specific comments: I agree with reviewer #1 that information is missing in the figures: Fig. 3: add data for BHT isomers in other cores (S5 for 64PE406 and S73 for ODP 160) Fig. 4: it would indeed be insightful to show ratios and SC ladderanes (see comments by reviewer #1). Drawing a line between points would also be helpful. The depth scale can be removed for the plot 4b (and generally, a and b are not needed).*

**We agree with Dr Blanchet and the anonymous reviewer who also brought up these points, and we will amend the revised manuscript to include these figure changes.**

*If possible, indicate the various sub-layers in the sapropel (proto-sapropel, oxidized sapropel) using the Ba and Mn concentrations (or as ratio over Al or Ti). Ba is a good indicator for sapropel extend and Mn shows the upper extend (upper redox front), so the oxidized part of the sapropel (where the TOC is low). If you have some time indication, it might be interesting to indicate/plot some results from other records (isorenioratene, forams, etc. . .) to get a fuller picture of the changes in water-column properties. Such a figure (depth profile) is missing for LC21, although a lot of data has been gathered on this core. This would allow direct comparison between other proxies and the anammox biomarkers, even at low sampling resolution.*

**To the best of our abilities we will include information about sapropel sections (as discussed above, using the accepted paper of Dirksen et al.) in the revised manuscript and the new figure. However, the low sampling resolution of the LC21 core makes it difficult to draw conclusions about anammox functioning within the sapropels. Rather all we can discuss is presence vs. absence instead of the timing or sequences of events. This was one of the main reasons we chose to study the high resolution 64PE406 core.**

*Fig. 5: please also connect dots with a line in 5b and if possible, indicate the various horizons in the sapropel (see comments for Fig. 4). While reading section 3.1, I was wondering why ladderanes had not been measured in 64PE406, and it is only when I read section 3.2 that I got my answer. It should be clear from the beginning that ladderanes were measured both in LC21 and 64PE406 (also in the method part, section 2.4.2) but that they could not be detected in the latter one.*

*Introduction line 48-54: perhaps introduce the meaning of anoxia vs. euxinia for nonspecialists? In general, it would be more accessible if terms would be better introduced (e.g., chemocline vs. redoxcline) or shown on schematic representations.*

*l. 365-371: I find this part quite obscure: what is meant by "Then, once monsoonal discharge brought in the initial pulse of nutrients from the Nile, [. . .]"? I do not follow the order of events. Perhaps making that appearing on fig. 4 would be helpful (e.g., by comparing to timing of freshwater pulses and development of anoxia)? Or draw small sketches? Similarly, with the proposal that the observed signal might be related to "split-anoxia": not very clear why that happens and might be useful to provide a visualization.*

**We will amend the MS to include these revision suggestions. A cartoon, as suggested in the earlier comment, will also better explain the order of anammox events in sapropels.**

*But once again, I enjoyed reading this paper and feel that it will contribute value to our understanding of changes in the marine environment related to deoxygenation processes, which were recently highlighted as a growing concern for present oceanic basins.*

---

## Author Response (AR2)

Dear S. Wajih A. Naqvi,

On behalf of the authors, I would like to thank you and Dr Blanchet for again taking the time to review this submission. We are pleased with the suggested technical correction suggestions and have updated Figure 3 to fit with Dr Blanchet's suggestion. We have also shortened the abstract as much as we feel is possible and removed many of the brackets that were included in the original.

Sincerely,

Darci Rush